# Cereal B-Glucans: The Impact of Processing and How It Affects Physiological Responses

**DOI:** 10.3390/nu11081729

**Published:** 2019-07-26

**Authors:** Muriel Henrion, Célia Francey, Kim-Anne Lê, Lisa Lamothe

**Affiliations:** 1Science & Technology Dairy, Nestle Research & Development Orbe, Route de Chavornay 3, CH-1350 Orbe, Switzerland; 2Institute of Health Sciences, Nestlé Research, Route du Jorat 57, CH-1000 Lausanne, Switzerland; 3Institute of Materials Science, Nestlé Research, Route du Jorat 57, CH-1000 Lausanne, Switzerland

**Keywords:** β-glucans, fibres, glucose, lipids, processing, viscosity

## Abstract

Cereal β-glucans are dietary fibres primarily found in oats and barley, and have several positive effects on health, including lowering the postprandial glucose response and the improvement of blood cholesterol levels. Cereal β-glucans have a specific combination of β-(1→4) and β-(1→3) linkages into linear long-chain polysaccharides of high molecular weight. Due to their particular structure, cereal β-glucans generate viscosity within the intestinal tract, which is thought to be the main mechanism of action responsible for their positive health effects. However, cereal grains are rarely consumed raw; at least one cooking step is generally required before they can be safely eaten. Cooking and processing methods more generally will modify the physicochemical characteristics of β-glucans, such as molecular weight, extractability and the resulting viscosity. Therefore, the health impact of β-glucans will depend not only on the dose administered, but also on the ways they are processed or converted into food products. This review aims at summarizing the different parameters that can affect β-glucans efficacy to improve glucose and lipid metabolism in humans.

## 1. Introduction

The consumption of dietary fibre is associated with the prevention of several chronic cardiometabolic diseases, including type 2 diabetes and cardiovascular diseases. For this reason, worldwide dietary guidelines tend to include high fibre intake in their recommendations. However, the daily recommended intake of 25 to 38 g of fibre for an adult does not include a specific guidance regarding the type of fibre to consume [1,2,3,4]. Indeed, there are many sources of dietary fibre in our diets, such as cereal grains, fruits and vegetables, nuts, seeds and legumes, and their physicochemical properties vary widely. Among the dietary fibre sources, cereal β-glucans have been shown to be one of the most effective fibre types to prevent type 2 diabetes and cardiovascular diseases. This is likely due to their ability to effectively lower the postprandial glucose response, as well as to improve long-term blood cholesterol levels, a known risk factor for cardiovascular diseases. 

Cereal β-glucans are linear homopolysaccharides composed of D-glucopyranosyl residues linked via a combination of β-(1→4) and β-(1→3) linkages [5]. Their structural pattern consists of consecutive β-(1→4) linked β-D-Glucose in blocks separated by a single β-(1→3) linkage, building a “staircase”-like structure, as illustrated in Figure 1. The resulting polysaccharide is built mainly of β-(1→3) linked cellotriosyl (58‒72%) and cellotetraosyl (20‒34%) units, but there is evidence of a small number of sequences with consecutive β-(1→4) linkages longer than tetraose and up to 14 glucosyl units [5,6]. Structural differences in cereal β-glucans are indicated by the trisaccharide to tetrasaccharide ratio and occur within different genera of cereals but not within the same genera. Cereal β-glucans are naturally present in oat and barley, each containing around 4.5% β-glucans [7]. These polymers also occur in rye (up to 2.5% in the whole flour) and in wheat, at lower concentrations (up to 2.5% in the bran fraction) [8,9]. Production of β-glucan-rich fractions from these raw materials is possible following multi-step extraction procedures [8,10].

The physicochemical properties of the β-glucan polymer have been extensively studied and reviewed in vitro by Wood et al. [11]. In their raw, native form, cereal β-glucans generate high viscosity, which is shear-sensitive at low concentrations. After a critical concentration is reached (the concentration at which sections of the polysaccharide begin to overlap), the viscosity increases exponentially with concentration and molecular weight (Mw). Viscosity increase is even greater at higher concentrations, which could be due to specific molecular associations or the presence of aggregated particles modifying the molecular entanglements [11]. 

β-glucans, whether arising from cereals, fungi or other sources, have attracted the attention of scientists and industry for more than two decades, and aspects varying from their extraction to their physiological impact and industrial relevance have been reviewed. Zhu et al. [12] published a thorough overview of their production means and industrial applications, including food but also cosmetic, medicinal and even feed usages. The specific usage of cereal β-glucans for functional food was discussed by Öning [13]. Lazaridou and Biliaderis [14] described all molecular aspects of cereal β-glucans in relation to their functionality, whilst Wang et al. [15] recently focused on the conformation of these molecules. Hu et al. described the structure and characteristics of cereal β-glucans in view of their extraction procedures [10]. State-of-the-art studies are available on their potential for glycaemic control in diabetes [16] or for the immune response [17,18]. Analytical methodologies to characterize β-glucans have also been thoroughly examined by some authors [19]. 

As previously mentioned, the physicochemical properties of cereal β-glucans, such as Mw, are the main determinants of their viscosity-generating properties. Therefore, it is important to pay close attention to these physicochemical properties when evaluating their impact on physiological responses. Furthermore, cereal β-glucans have usually undergone some level of cooking or processing before being consumed, so an understanding of how their physicochemical properties change during cooking or processing is paramount. In this review, we take a fresh perspective by discussing the differences observed in physiological responses according to Mw and the extractability of cereal β-glucans as well as the impact of the food matrix. In addition, the impact of several processing techniques on β-glucans’ physicochemical features are discussed in light of their physiological impact. 

## 2. The Effect of β-Glucans on Glucose Metabolism

The positive effects of β-glucans on blood glucose are well established. For instance, the European Food Safety Authority (EFSA) approved a health claim for lowering glucose response when at least 4 g of β-glucans, from oat or barley, per 30 g of available carbohydrates are consumed in a meal [20]. The postulated mechanism of action is that, due to the viscosity they generate, cereal β-glucans will result in a slower rate of gastric emptying, which, in turn, delays the delivery of the chyme to the intestine. In addition, as cereal β-glucans are resistant to digestion, they remain intact in the small intestine and increase the viscosity of the alimentary bolus. The increased viscosity will cause a delay in the digestion and absorption of available carbohydrates due to several mechanisms: a) a thick unstirred layer around the food bolus that hinders access of intestinal enzymes to substrates, b) a lower mixing of the luminal content, and c) retarded transport of glucose to the absorbing surface [20,21]. Then, following a lower and slower appearance of glucose into the blood stream, insulin concentrations are also reduced [22]. While the dose of cereal β-glucan is supposed to be the primary factor that influences these mechanisms, other parameters, such as Mw and extractability, will affect these responses. 

### 2.1. The Effect of Molecular Weight of Cereal β-Glucans on Glucose Response

Cooking or industrial processing of cereal β-glucans has an impact on their physicochemical characteristics. Due to the high viscosity that cereal β-glucans generate, the structural changes that they undergo during processing allow for improved processability and sensory properties, but also influence their physiological benefits. Structural changes include polymer size (related to Mw) and the amount of polymer that is released from its food matrix (related to extractability). If both aspects are important in viscosity development, they have nevertheless often been studied separately. Many studies, mostly in healthy subjects, have investigated the effect of cereal β-glucans with different Mw on glycaemic response [23,24,25,26,27,28]. It is generally concluded that cereal β-glucans with higher Mw have a greater impact on the reduction of postprandial blood glucose response (PBGR), area under the curve (AUC), and glycaemic index of beverages and cereal-based products [23,24,26]. Although this general observation is consistent, the ranges in Mw that were tested vary widely and the results are confounded by factors such as the dose and type of food matrix. For instance, in a study looking at the incorporation of 4 g of low- (145 kDa) or high- (580 kDa) Mw β-glucans in a 50 g glucose solution, no significant decrease of glucose AUC was observed compared to the glucose control drink [23]. However, the PBGR curve was broader and more flattened with the solution prepared with 580 kDa β-glucans. In another study testing a liquid food format, Thondre et al. supplemented a soup with either low- (150 kDa) or high- (650 kDa) Mw β-glucans and observed a significant reduction of the PBGR only after consumption of the soup prepared with β-glucans of 650 kDa, in comparison to the control soup without β-glucans [28]. However, the soup prepared with 650 kDa β-glucans contained 3.2 g of β-glucans, while the soup prepared with 150 kDa β-glucans contained a lower amount, 2.7 g. This difference in β-glucan content makes it difficult to come to any conclusion about the impact of Mw. In the case of solid food formats, the impact of Mw is also inconsistent. A dose of 6.2‒6.3 g β-glucans per 40 or 60 g of available carbohydrates was provided by granola [25]. The Mw of β-glucans tested ranged from 2133 kDa to 32 kDa and, in comparison to the control granola product, only β-glucans of 2133 kDa were significantly more effective at reducing the glucose iAUC and the PBGR than other granola formulations with β-glucans ranging from 32 kDa to 435 kDa. A further study by Thondre et al. showed that low-Mw, high-purity β-glucans extracted from barley had no significant effect on glucose release during in vitro digestion or on the glycaemic response elicited by chapatti bread samples in which they had been incorporated. The authors concluded that low-Mw barley β-glucans were ineffective at lowering the glycaemic response, possibly due to their inability to hinder starch digestion and increase gastrointestinal viscosity after consumption of the bread [27]. Unfortunately, the authors failed to provide the Mw values for each source of β-glucans tested, as is the case for many studies reporting on their glycaemic-lowering effect. Therefore, it is difficult to determine the optimal Mw range to elicit a significant effect or how the food matrix may influence the result. Furthermore, there are considerable differences in the Mw data reported in the literature for the same type of β-glucans. Extraction conditions, partial depolymerization by endogenous enzymes and differences in analytical techniques for Mw determination contribute to such variability. 

### 2.2. The Effect of Cereal β-Glucans’ Extractability on Glucose Response

As previously discussed, the viscosity of β-glucans depends on the concentration and Mw, but the solubility of this compound, both in the food matrix and digesta, is also a determinant of its glycaemic-lowering impact [29]. Indeed, β-glucan solubility is likely to affect its viscosity-generating properties. Recent work has demonstrated that β-glucan solubility in foods depends on the source of the material and processing conditions, such as fermentation and freezing, and is subject to changes during storage [30,31]. Furthermore, the form in which the β-glucans are present within the food product influences the glycaemic response to it, as shown by Kwong et al. [23]. These authors reported that viscous solutions containing β-glucans elicited a lower glycaemic response than gels made out of the same amount and type of β-glucans. These results indicate that the ability of increasing digesta viscosity is determined by the ability of β-glucans to trap liquids in the gastrointestinal lumen. If β-glucans are not free to interact with luminal fluids, as is the case when they are in gel form, their glycaemic-lowering impact is reduced. These types of results highlight the importance of analysing foods containing β-glucans under physiological conditions so that product development targets the maximum functionality and physiological impact. 

In a study where β-glucans were incorporated into muffins that were subjected to freeze‒thaw cycles, the PBGR was not significantly reduced in comparison to control muffins that did not contain β-glucans. It was observed that freeze‒thaw treatments reduced the solubility of β-glucans, which attenuated their physiological effectiveness in reducing the postprandial glycaemic response [32]. This processing-induced change will be discussed in a further section. Gamel et al. reported on the impact of amylase, protease and lipase digestions on the solubility and resulting viscosity of β-glucans from oat bran cereals. Their results showed that enzymatic digestion of the cereals that contained high-Mw β-glucans increased the final viscosity by facilitating their release from the food matrix. On the other hand, in cereals that contained partially depolymerized β-glucans, digestive enzymes decreased their final viscosity by removing starch and protein. Furthermore, the viscosity varied depending on the enzyme combinations, which, in vivo, would change according to the composition of the food consumed. The viscosity of β-glucans digested with pancreatin plus microbial α-amylase (pH 6.9) was predictive of LDL-cholesterol reduction (*R^2^* = 0.847) and glycaemic response (*R^2^* = 0.883) [33]. It appears that, in order to generate viscosity and have a significant lowering impact on the glycaemic response, β-glucans need to be readily solubilized. However, it becomes increasingly clear that this solubilization and subsequent generation of viscosity is only relevant if it can occur in the gastrointestinal lumen. For this reason, the availability or release of β-glucans during digestion of the test sample should be accounted for when analysing the glycaemic-lowering property of β-glucans within a specific food matrix.

### 2.3. The Effect of Food Matrix on the Impact of Cereal β-Glucans on Glucose Response

Most data on glycaemic response following the consumption of β-glucans come from studies involving healthy subjects but using different sorts of food matrices, such as bread, spaghetti, cookies, cereals, porridge and beverages [34,35,36,37,38,39]. Without considering Mw, different effects on glycaemic response are observed for similar doses of β-glucans. Souki-Rincón et al. compared the consumption of traditional arepas from Latin America, a food made of ground maize dough, to arepas supplemented with 4.2 g of β-glucans per 50 g of available carbohydrates [39]. The intervention significantly decreased PBGR without leading to increased insulin secretion. The same amount of β-glucans in a muesli consumed with yoghurt significantly lowered glucose and insulin responses compared to the same meal without β-glucans, while a lower amount (3 g) of β-glucans did not achieve a significant effect in this study [37]. Another type of matrix, a porridge from wheat flour, was used to incorporate 8.8 g of β-glucans per 30 g of available carbohydrates in a study by Braten et al., which resulted in a significantly lower glycaemic response in healthy subjects, as well as in type 2 diabetic subjects [35]. Similarly, the addition of β-glucans to barley cookies (3.5 g per 40 g available carbohydrates) significantly decreased postprandial glucose and insulin response compared to whole-wheat cookies. However, this effect was not observed with crackers (3.6 g per 40 g available carbohydrates) as compared to their whole-wheat control [36]. Also, in subjects presenting with metabolic syndrome, 4.2 g of β-glucans per 50 g of available carbohydrates in wheat bread had a positive effect on postprandial glucose compared to wheat bread and on postprandial insulin compared to wheat bread with arabinoxylan [38]. When added to a liquid matrix, a higher concentration of β-glucans was tested (5 g per 50 g of available carbohydrates) in a beverage given to a population of hypercholesterolemic subjects, with similar positive results on postprandial glucose and insulin compared to a rice-starch-enriched beverage. However, this observation was valid for oat β-glucans but not for barley β-glucans [34]. Thus, it appears from these various studies that the effective dose of β-glucans may depend on the type of food matrix. This subset of studies indicates that solid foods may require lower doses of β-glucans than liquid or semi-solid foods. 

## 3. The Effect of β-Glucans on Lipid Metabolism

The effect of cereal β-glucans on blood lipids has not been investigated as thoroughly as their effect on glucose metabolism. One of the reasons for this may be that the cholesterol-lowering effects of β-glucans occur after a longer period (5‒6 weeks, vs. acute effect for glucose-lowering effect) and this makes such type of studies difficult to control for subject compliance. The EFSA panel reviewed the evidence regarding the cholesterol-lowering effects of cereal β-glucans and concluded that at least 3 g of oat β-glucans per day are required to claim a reduction of blood cholesterol in normo- or hypercholesterolemic adults, which results in a lower risk of (coronary) heart disease [40]. Similarly, the U.S. Food and Drug Administration (FDA) approved the claim that the use of at least 3 g of β-glucans from oat or barley per day leads to a reduction in coronary heart disease resulting from lower circulating cholesterol levels [41]. The consumption of β-glucans increases bile acid excretion, which in turn stimulates the metabolism and elimination of cholesterol. The measurement of markers of cholesterol synthesis in the blood indicates a higher elimination and lower absorption of cholesterol [42]. Additionally, an increased viscosity caused by the presence of β-glucans would decrease the absorption of cholesterol.

### 3.1. The Effect of Molecular Weight of Cereal β-Glucans on Lipid Response

Studies looking at the chronic effects of β-glucan supplementation in the diets of mildly hypercholesterolemic subjects have tested doses of 2‒6 g per day in both solid and liquid food matrices [43,44,45,46]. Two studies investigated the effect of barley β-glucans of different Mw, 290 kDa and 1350 kDa, in two different doses, 3 and 5 g [44,45]. In the first study, where β-glucans were included in ready-to-eat cereal and fruit juice, both high and low MW were efficient at reducing LDL- and total cholesterol after six weeks of consumption, as compared to the control products without β-glucans. The second study provided five weeks of a breakfast made of crepes, tortillas, porridge or chips formulated with β-glucans. A significant reduction of circulating total cholesterol was observed only with high-Mw β-glucans, as compared to the control products formulated with wheat and rice. They concluded that the physicochemical properties (i.e., Mw) rather than the daily intake of β-glucans are determining factors for the cholesterol-lowering effects. Two other studies investigated the supplementation with β-glucans from oat [43,46]. Wolever et al. noticed a significant reduction in LDL-cholesterol after four weeks of diet supplementation with ready-to-eat breakfast cereals containing 3 or 4 g of either medium- (850 kDa) or high-Mw (2250 kDa) β-glucans compared to wheat cereals [46]. The low-Mw (530 kDa) β-glucans did not lead to a significant effect. In agreement with the findings of Wolever et al., Frank et al. did not observe any significant change related to chronic consumption of 5‒6 g of β-glucans with 217 kDa or 797 kDa Mw [43]. These studies indicate that, as opposed to glycaemic response, the effect of cereal β-glucans on blood lipids appears to be more consistently dependent on Mw, as was also reported in a review investigating the effects of β-glucans on cholesterol [47].

### 3.2. The Effect of Food Matrix of Cereal β-Glucans on Lipid Response

Studies investigating the effect of β-glucans on lipid parameters mostly included hypercholesterolemic subjects, although often only with mild hypercholesterolemia, and looked at the results of two to eight weeks’ consumption on blood cholesterol and triglyceride levels as compared to the same diet without supplementation. Commonly, interventions provided around 5‒6 g of β-glucans per day, included in various types of food matrices, such as pasta, bread, cake, soups, cookie, sauce or orange juice [34,48,49,50,51]. Theuwissen et al. [51] incorporated 5 g of β-glucans in a wheat muesli consumed as part of a regular diet for four weeks, which significantly lowered serum total and LDL-cholesterol through the increase of bile acid synthesis. In the study of Rondanelli et al., the addition of β-glucans into various products of the diet significantly decreased LDL- and total cholesterol in the blood after four weeks [50], while in the study of Kerckhoffs et al., only β-glucans added to orange juice, but not bread or cookies, caused a significant decrease in the same metabolic parameters [48]. A further study investigating the addition of 5 or 10 g of β-glucans in a beverage, complementing the usual diet for eight weeks, observed a significant decrease of total cholesterol concentrations with β-glucans from oats but not from barley [34]. Moreover, this study noticed that the amount of β-glucans incorporated into the food does not necessarily predict the magnitude of the effect on serum cholesterol. Naumann et al. observed significant changes in the serum total and LDL-cholesterol concentrations when 5 g of β-glucans were consumed for five weeks as a daily fruit drink [49]. A review studying the impact of β-glucans on lipid metabolism included a wide range of studies using β-glucans or oat ingredients in liquid, semi-solid and solid matrices. They concluded that with liquid oat-based foods, cholesterol reductions were moderate but more consistent than with solid or semi-solid foods with similar amounts of β-glucans. However, the results were more conflicting when the matrix of consumption was more complex [47]. It may be highlighted that addition of β-glucan in beverages (e.g., oat extract), especially acidic ones such as fruit-based beverages, might result in a reduction of the β-glucans Mw [52]. The mechanisms involved seems to be related to oxidative cleavage rather than acid hydrolysis, and degradation may also take place during storage of the beverages [53].

## 4. The Impact of Cooking and Industrial Processing on the Physicochemical Properties of Cereal β-Glucans

### 4.1. Breadmaking and Preparations with Enzymes

High viscosity is a hallmark of β-glucans, and is a function of both concentration of the β-glucans in solution and their Mw. As the resulting viscosity development follows an exponential curve, any impact on either one of these parameters will largely impact the rheological behaviour of the β-glucan matrix studied [11]. Several studies describe the impact of processing on both extractability and Mw of cereal β-glucans. It is worth highlighting that measurements of β-glucan extractability are dependent on many factors including temperature, pH and presence of digestive enzymes. For example, increasing extraction temperature increases the solubility of the polymer, which results in greater extraction. Therefore, in vitro extractability procedures designed to mimic gastrointestinal conditions, as the Beer method [54], will give much lower values than traditional hot water extractions. 

Breadmaking is a very popular way to eat cereal grains, however, breads are predominantly produced from wheat. Amount of β-glucans in bread can nevertheless be increased through either use of flours with higher levels of β-glucans (e.g., in rye breads) or through addition of β-glucan-rich fractions (e.g., wheat-based breads enriched in oat bran). Endogenous enzymes naturally present in wheat, but also in β-glucan-containing flours, are susceptible to greatly modify the β-glucan polymer structure [55]. This specific topic was thoroughly studied by Andersson et al. in oat-bran-enriched rye breads using rye flours with varying endogenous enzyme activity [56]. The authors reported a systematic decrease in Mw with fermentation time within the first few minutes of fermentation. Interestingly, endogenous enzymes were concomitantly responsible for an increased extractability of β-glucan polymers through degradation of other cell wall polysaccharides that are naturally entangled with β-glucans in the grain. With short fermentation times, extractability doubled as compared to native flour but then decreased at long fermentation times below native extractability. This latter effect was related to the self-association of β-glucan oligomers to form insoluble complexes. The level of endogenous enzyme activity itself did not affect the content in water-extractable β-glucans. Finally, oven baking had only a negligible impact on both β-glucans Mw and extractability. Several authors have reported both polymer degradation [57,58,59] and improved extractability [59,60] during breadmaking. The resulting effect of bread processing on β-glucan-related viscous behaviour will thus be highly dependent on the fermentation process. A recent study highlights that some barley cultivars are more resistant to these fermentation-related changes, bringing interesting perspectives to the manufacturing of β-glucan enriched breads [59]. 

The self-aggregation of enzymatically generated β-glucan oligomers has also been reported by Tosh et al. on muffin preparation including β-glucanase [61]. The partial depolymerization resulting from the enzyme addition affected extractability of β-glucans to a varying extent: the extractability increased with decreasing Mw then declined following the self-assembly of gel networks in solutions, arising from the partially depolymerized β-glucans. The lowest extractability values were measured at 17% at low Mw versus 54% at high Mw. As previously mentioned for bread, the baking of treated muffins (30‒60 min at 180 °C) itself did not further impact the β-glucans. 

### 4.2. Domestic Cooking and Storage of β-Glucan Formulations

If the baking of a pre-fermented dough does not appear to further impact the state of β-glucans, cooking itself is nonetheless likely to release β-glucans from the cell wall matrix into which they are embedded, thus increasing β-glucans’ extractability. This, however, often occurs without any drastic changes in Mw. Typically, oat bran, cooked porridge, muffins and breakfast cereals are foods that have been shown to contain high-Mw β-glucans [52]. The physical format of the β-glucan ingredient may modulate the cooking impact as well. As an illustration, Beer et al. [54] observed that cooking oat bran into porridge did not affect the extractability of β-glucans (as measured under physiological conditions), but that a slight increase in extractability was observed when cooking rolled oats. Johansson et al. [62] also report that the addition of oat flakes to boiling water followed by 10 min cooking generated an increased amount of soluble β-glucans (extracted from soluble fibre fraction). Similarly, Aman et al. [52] reported that the use of a large particle size of bran, and a short fermentation time, can limit the β-glucan degradation observed during breadmaking. Aside from the ingredient format, the whole recipe including the β-glucans may impact the cooking effect. As observed by Beer et al. [54], the muffin-baking process (20 min at 200 °C) led to lower Mw but an increase in β-glucans’ extractability. The intensity of the effects depended greatly, however, on the muffin recipe. 

Submitting cooked foods into freeze‒thaw cycles may also impact β-glucans’ extractability through molecular rearrangement. As reported by Beer et al. [54], freezing storage decreased β-glucans’ extractability in muffins but did not affect Mw. After eight weeks of freezing storage, the extractability was reduced by 50%. The authors attributed these changes to molecular organization during storage (i.e., chain aggregation), leading to water being repelled from the polysaccharide matrix. When submitting muffins to several freeze‒thaw cycles, Lan-Pidhainy et al. [32] also observed decreased extractability, from 24% after two cycles up to more than 50% after four cycles. Although the Mw after four cycles was significantly lower, all muffins kept overall high Mw values.

### 4.3. Extrusion Cooking

Aside from muffins and breadmaking, a popular way of processing β-glucans-rich foods is extrusion. Extrusion may be distinguished from traditional porridge cooking or dough baking by the high pressure/high shear that will be applied to the product. Extrusion may impact both β-glucan Mw and extractability depending on the parameters applied. When processing two different barley cultivars (waxy and regular) using twin-screw extrusion at different moisture levels (20‒50% moisture) and different temperatures (90, 100, 120 and 140 °C), authors Gaosong and Vasanthan [63] measured increased extractability that was both cultivar- and process-parameter-dependent. For the waxy sample, solubility decreased with increasing moisture level at each extrusion temperature tested. The regular cultivar showed higher extractability after extrusion and increased extractability with increasing moisture at each temperature level. Differences were attributed to cultivar-dependent primary structural differences. The waxy cultivar also showed some fragmentation during extrusion, more pronounced over 120 °C. The regular cultivar seemed to be rather resistant to extrusion. Tosh et al. [64] extruded full recipes of oat bran, corn flour, fructose and salt under different conditions including moisture, temperature and specific mechanical energy applied (SME). They measured β-glucan features (extractability and Mw) after using an in vitro digestion method using human salivary α-amylase, porcine pepsin and pancreatin. Authors report exponential relationship between SME x Temperature and β-glucan Mw in the finished product (under physiological extraction). Increasing temperature and SME markedly increased the extractability of β-glucans from 67% to 100%. The concentration of β-glucans in the extracts increased with increasing temperature and SME, but the viscosity decreased due to the concurrent decrease in Mw. Increasing temperature and SME and decreasing the water added to the extruder resulted in depolymerization of the β-glucans. Neither high- nor low-Mw β-glucans were preferably extracted in these oat cereals using the physiological extraction protocol. Using oat bran alone, Zhang et al. [65] also reported a significant impact of the extrusion parameters (feed moisture and temperature) on the soluble dietary fibre content (mostly made from extractible β-glucans). An extractability increase was also reported on hulled barley husks (up to 8% at high temperature–low moisture extrusion conditions) [66].

Traditional processing of β-glucan-rich foods will thus invariably lead to a change in either Mw (polysaccharide breakdown) or extractability (release from cell wall or aggregation) or both, as summarized in Table 1. These changes are highly process-dependent and, even if some general trends arise, they are difficult to predict. Furthermore, the physical source of β-glucans, bran, flour or rolled grain, or the nature of the recipe that will be processed will also affect the structural change brought by processing. Changes in extractability and Mw will in turn affect the viscous behaviour of the food and both its palatability and the physiological effect that it may exert. Combining clinical approaches with a thorough characterization of the structure of the β-glucan polymers is therefore a critical aspect to consider.

## 5. Conclusions

The impact of β-glucans on blood glucose and cholesterol lowering has received much attention in the scientific community, leading to the approval of health claims by various health authorities. β-glucans, being naturally found in cereals, represent an attractive opportunity for food manufacturers to propose foods with added health value, while still bearing consumer-friendly labelling. Cereals are not usually consumed raw; at least one cooking step is generally needed before consumption. As summarized in Figure 2, any processing method will affect the Mw and/or extractability of β-glucans, which will in turn modulate their ability to achieve the expected health benefits within a given dose. Most intervention studies have provided extensive details on the dose of β-glucans tested, but information on food source, process and β-glucans Mw and extractability is often lacking. Generating more high-quality clinical interventions with proper technical evaluation of the food products and their behaviour upon digestion will allow us to better predict how specific processes will affect gastric and intestinal viscosity, and thus the expected effect on health. This would also allow for identifying the key parameters and their acceptable threshold to reach Mw—i.e., using the product of Mw × concentration—to accurately predict the effect of a β-glucan-containing food product on physiological parameters such as postprandial glucose response. Having a more systematic characterization of food products in clinical trials would require us to develop accurate analytical procedures reflecting β-glucans’ physiological behaviour upon digestion [54,61]. 

While this review has focused mostly on the well-characterized effect of β-glucans on glucose and lipid metabolism, novel research suggests that they may also play a role in a wide range of less-investigated health effects. The ability of β-glucans to form a highly viscous solution in the gastrointestinal tract and to delay the absorption of nutrients might potentiate the feeling of satiety and decrease subsequent energy intake. However, several studies using β-glucans conclude differently [67,68,69,70,71]. Generally, a higher consumption of β-glucans (>5 g per meal) seems to induce higher satiety [67,68] than a lower amount (<5 g per meal) [69,70,71]. Both solid and liquid foods have a positive impact on satiety, but a study showed greater effectiveness with the consumption of beverages [68]. β-glucans also exert favourable effects on intestinal function and gut health through an activity on intestinal microflora and bacterial metabolites [72]. Additionally, β-glucans from bacterial and fungal origin have recognized benefits for the stimulation of the immune system, as antimicrobial agents and against tumour and metastasis development, and they are used as prevention treatment, as well as in adjuvant therapy [17]. Some evidence also attributes such positive effects to β-glucans from cereal sources. Several in vitro studies highlighted the anticancer and pro-apoptotic properties oat β-glucans [72,73,74,75], while barley β-glucans displayed an antimutagenic effect [76,77,78]. In contrast to their effect on glucose and lipid metabolism, low-Mw β-glucans are more effective at inducing the expression of a marker of cell apoptosis in cancer cell lines in comparison to normal human and murine cell lines, and therefore have more notable anticancer properties [73]. Moreover, oat β-glucans proved efficient against microbial and parasite infection [79,80]. Finally, skin health also seems to be impacted by the consumption of β-glucans carrying antioxidant properties and anti-wrinkle activity. Indeed, cosmeceutical applications incorporating oat β-glucans already exist and are used for their soothing, moisturizing and anti-irritant properties [81]. Evidence on these health benefits is still scarce, especially human data and research on the impact of β-glucans’ processing. Further studies are needed to determine the impact of β-glucans consumption on these different health benefits. 

In addition to health outcomes, an important parameter to consider is how changes in β-glucans Mw will affect the palatability and sensory properties of the products. In order to make a cereal product palatable, most processes affect these properties, precisely in order to decrease the viscosity. Novel research aiming at identifying the key characteristics required for β-glucans to achieve their desired effect will help guide the development of healthy yet palatable food products. 

## Figures and Tables

**Figure 1 nutrients-11-01729-f001:**
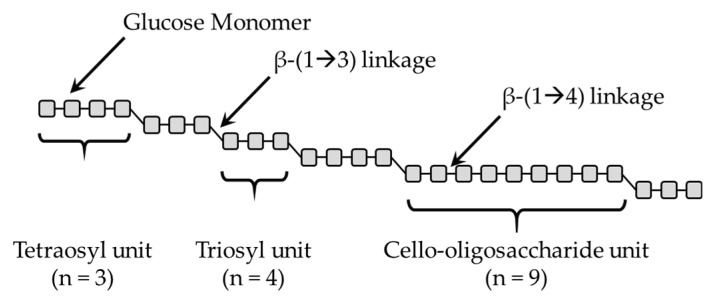
Cereal β-glucans structure, adapted from Tosh et al. [6].

**Figure 2 nutrients-11-01729-f002:**
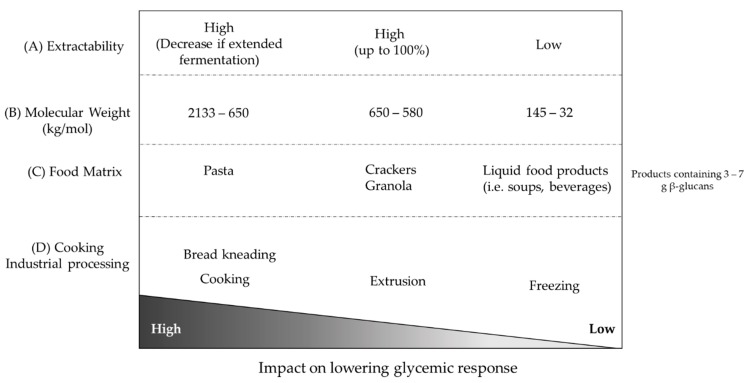
A schematic overview of the typical processing-induced changes to cereal β-glucan structure reported in this review. The ability to lower glycaemic response is expressed as a gradient from high to low. (**A**) A high β-glucan extractability has more effect on decreasing glucose response as compared to a low extractability. (**B**) Similarly, β-glucans with increasing molecular weight have better ability to reduce glucose response. (**C**) Pasta with 3‒7 g of β-glucans shows the highest efficacy for decreasing glucose response, while crackers and granola have an intermediate effect. Liquid food products, such as soups and beverages, have a low impact on the reduction of glucose response. (**D**) The processes of cooking and bread kneading retain β-glucans’ ability to lower glycaemic response, extrusion decreases this ability, and β-glucans undergoing the process of freezing have a low ability to decrease glucose response.

**Table 1 nutrients-11-01729-t001:** Overview of typical processing-induced changes in cereal β-glucan structure.

Type of Processing	Effect of Processing
Enzymatic Processing (e.g., Breadmaking)	Large Mw reduction
	Increased extractability
	Gelling domain reached
Cooking & Extrusion	Large Mw reduction under extreme cooking conditions (i.e., low moisture, high heat)
	Increased extractability
Freezing & Freeze‒Thaw Cycles	Decreased extractability due to aggregation

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
