# Peer review of "Cereal B-Glucans: The Impact of Processing and How It Affects Physiological Responses"

_nutrients, 2019, doi:10.3390/nu11081729_

Reviewer 1 Report

The manuscript "Cereal b-glucans: impact" is a review on the effect of processing on physiological responses such as glucose/lipid metabolisms and response, of b-glucan extractability on glucose response, and on physicochemical properties of cereal b-glucans.

The Review deals with an interesting topic and is well organised and easy to follow. However, my suggestion is to add one table summing up the key aspects analysed and discussed within the text to support and improve the paper readibility. Paragraph 4 “Impact of cooking and industrial…” might also be splitted in subparagraphs.

Few aspects to revise and amend, please:

-       Add some Reference(s) in lines 86, 152, 232.

-       Few minor English mistakes to correct, such as using the pronoun “their” instead of “its” when b-glucans are referred to (e.g., line 33, etc.), “trough” in line 218, etc.

-       Authors within the text are indicated as “et al.” and “ and coworkers”. Please, check Journal guidelines and make the choice consistent within the text.

-       In a few cases b-glucans are with “b” instead of “β”. Please, make consistent.

-       Check Figure 1 font.

Reviewer 2 Report

GENERAL COMENTS:

The manuscript presents very interesting review about β-glucans. A great interest of β-glucans with many health-promoting and prebiotic properties has been registered and this work is valuable but in my opinion some aspects should be detailed.  In my opinion authors focus on the well-known aspects of the impact of beta glucan on glucose and lipid metabolism. Maybe this should be mentioned in the summary which other aspects of health promoting should be investigated for example beta glucans can stimulate immune functions by activating monocytes/macrophages and increasing the amounts of immunoglobulin, NK cells, killer T-cells, and so on, which will improve resistance to cancer and infectious and parasitic diseases.  

INTRODUCTION SECTION

Introduction section should be improved

Page 1, line 35

After sentence

 Cereal β-glucans are naturally present in oat and barley, each containing around 4.5% β-glucan it should be mentioned that the high purity wheat β-d-glucan was also obtained from white wheat bran after alkali extraction and multi-precipitation with ammonium sulphate (Li et al. 2006).

Before last sentence …… In this review, we would like to discuss the differences in physiological responses ……………  Authors should cite other review studies which have been already done about similar topic regarding β-glucans and explain what new issues are addressed in this study. For example Lazaridou and Biliaderis (2007) mentioned molecular aspects of cereal beta-glucan functionality. Zhu et al. (2016) presented a critical review on production and industrial applications of beta-glucans. Hu et al. (2015) described structure and characteristic of beta-glucan in cereal. These references should be mentioned in this work

Effect of food matrix of cereal β-glucans on lipid response

Page 5, Lines 210-227

Line 227.

After sentence “Naumann et al. observed significant changes in the serum total- and LDL-cholesterol concentrations when 5 g of β-glucans were consumed for 5 weeks as a daily fruit drink” authors should add  that a review of the literature made by Grundy et al. (2018) suggested that for a similar dose of β-glucan, liquid oat-based foods seem to give more consistent, but moderate reductions in cholesterol than semi-solid or solid foods where the results are more variable

Although reduction of LDL cholesterol is the main adventage after consumption of beta-glucans it should be also mentioned about interesting studies that enrichment of biscuits and juice with oat β-glucan enhances postprandial satiety (Pentikäinen et al. 2014).

Conclusion

Conclusion section is too long it should be more specific, and show research trends and deficiencies in the available literature on the topic

Grundy, M. M. L., Fardet, A., Tosh, S. M., Rich, G. T., &      Wilde, P. J. (2018, March 1). Processing of oat: The impact on oat’s      cholesterol lowering effect. Food and Function. Royal Society of      Chemistry. https://doi.org/10.1039/c7fo02006f.

Hu, X., Zhao, J., Zhao, Q., & Zheng, J. (2015). Structure      and Characteristic of β-Glucan in Cereal: A Review. Journal of Food      Processing and Preservation, 39(6), 3145–3153.      https://doi.org/10.1111/jfpp.12384

Lazaridou, A., & Biliaderis, C. G. (2007, September).      Molecular aspects of cereal β-glucan functionality: Physical properties,      technological applications and physiological effects. Journal of Cereal      Science. https://doi.org/10.1016/j.jcs.2007.05.003.

Li, W., Cui, S. W., & Kakuda, Y. (2006). Extraction, fractionation, structural and physical      characterization of wheat β-d-glucans. Carbohydrate Polymers, 63(3),      408–416. https://doi.org/10.1016/j.carbpol.2005.09.025.

Pentikäinen, S., Karhunen, L., Flander, L., Katina, K.,      Meynier, A., Aymard, P., … Poutanen, K. (2014). Enrichment of biscuits and      juice with oat β-glucan enhances postprandial satiety. Appetite, 75,      150–156. https://doi.org/10.1016/j.appet.2014.01.002

Zhu, F., Du, B., & Xu, B. (2016, January 1). A critical      review on production and industrial applications of beta-glucans. Food      Hydrocolloids. Elsevier. https://doi.org/10.1016/j.foodhyd.2015.07.003

Author Response

Round  2

Reviewer 1 Report

Dear Authors,

thank you for amending the manuscript according to suggestions.

Please, mind line 252. English correction is required (subject and verb accordance).